# LANGUAGE MODEL COMPRESSION WITH WEIGHTED LOW-RANK FACTORIZATION

Yen-Chang Hsu[*1], Ting Hua[*1], Sung-En Chang[2], Qian Lou[1], Yilin Shen[1], and Hongxia Jin[1]

[1]Samsung Research America , [2]Northeastern University ,

{yenchang.hsu, ting.hua, qian.lou, yilin.shen, hongxia.jin}@samsung.com,{chang.sun}@northeastern.edu

## ABSTRACT

Factorizing a large matrix into small matrices is a popular strategy for model compression. Singular value decomposition (SVD) plays a vital role in this compression strategy, approximating a learned matrix with fewer parameters. However, SVD minimizes the squared error toward reconstructing the original matrix without gauging the importance of the parameters, potentially giving a larger reconstruction error for those who affect the task accuracy more. In other words, the optimization objective of SVD is not aligned with the trained model's task accuracy. We analyze this previously unexplored problem, make observations, and address it by introducing Fisher information to weigh the importance of parameters affecting the model prediction. This idea leads to our method: Fisher-Weighted SVD (FWSVD). Although the factorized matrices from our approach do not result in smaller reconstruction errors, we find that our resulting task accuracy is much closer to the original model's performance. We perform analysis with the transformer-based language models, showing our weighted SVD largely alleviates the mismatched optimization objectives and can maintain model performance with a higher compression rate. Our method can directly compress a task-specific model while achieving better performance than other compact model strategies requiring expensive model pre-training. Moreover, the evaluation of compressing an already compact model shows our method can further reduce 9% to 30% parameters with an insignificant impact on task accuracy.

## 1 INTRODUCTION

Language models built with transformers (Devlin et al., 2018) have attained extensive success in natural language tasks such as language modeling (Radford et al., 2018), text classification (Wang et al., 2018), question answering (Rajpurkar et al., 2016), and summarization (Liu, 2019). The success is achieved by fine-tuning a big transformer model pre-trained with a large corpus. The target task for fine-tuning may only focus on a restricted scenario such as sentiment analysis (Socher et al., 2013) and multiple-choice question inference (Zellers et al., 2018). Having a big transformer model is often overkill for the target task and prohibits the model deployment to resource-constrained hardware. Therefore, language model compression raises immense interest.

The popular strategy creates a compact model from scratch (Jiao et al., 2019) or a subset of the big model's layers (Sun et al., 2019; Sanh et al., 2019), then pre-trains with a large corpus and distills knowledge from the big model. This process is called generic pre-training (Wang et al., 2020b; Sun et al., 2019; Sanh et al., 2019) and is necessary for a compact model to achieve good performance on the target tasks. However, the generic pre-training could still cost considerable computational resources. For example, it takes 384 NVIDIA V100 GPU hours to get the pre-trained TinyBERT (Jiao et al., 2019) on the Wiki corpus dataset. So it may not be affordable for everyone who wants to create a compact model. In contrast, another line of strategy, specifically low-rank factorization (Golub & Reinsch, 1971; Noach & Goldberg, 2020), can potentially reduce a big model's parameters

---

[*]These authors contributed equally to this work.

without the generic pre-training. Since the factorization aims to approximate the learned model parameters, the method has the nature of directly inheriting the knowledge of the big trained model.

However, approximating the learned weights with standard factorization often loses most of the task performance. This work investigates this issue with the most popular strategy, which uses singular value decomposition (SVD) to compress the learned model weights. With SVD, the learned matrix is factorized into three matrices ($U$, $S$, $V$). The portion associated with small singular values will be truncated to produce a smaller version of factorized matrices. The multiplication of these smaller matrices will approximate the original one with fewer total parameters to achieve the model compression. In other words, SVD minimizes the reconstruction error with fewer parameters as its objective. However, this objective does not necessarily correlate to the ultimate goal of keeping task performance. Specifically, the SVD algorithm is biased to reconstruct the parameters associated with large singular values. As a result, the parameters mainly reconstructed by the ranks with small singular values will become the victim in the compression process. Are these victimized parameters less critical to achieving a good task performance? We argue that this is not true, and the optimization objective of SVD is not properly aligned with the target task objective. This paper is the first work to provide an empirical analysis of this issue, proposing a novel weighted SVD to mitigate it.

Our weighted SVD addresses the above issue by assigning importance scores to the parameters. This score has to correlate to how much the task performance is affected by the parameter change. The Fisher information nicely fits into this purpose (Pascanu & Bengio, 2014). Besides, the calculation of Fisher information is usually simplified to accumulating a parameter's squared gradient over the training dataset based on its task objective (*e.g.*cross-entropy, regression error, etc.), conveniently providing the importance of each parameter in a model. Then we modify the optimization objective of factorization (*i.e.*, reconstruction error) by multiplying it with Fisher information, providing a new objective that jointly considers matrix reconstruction error and the target task objective.

In summary, this work makes the following contributions: (1) we analyze the issue of mismatched objectives between factorization and the target task for model compression; (2) we propose a novel compression strategy with the SVD weighted by the Fisher information; (3) we perform extensive analysis on varied language tasks, showing our Fisher-weighted SVD can compress an already compact model, and it can achieve comparable compression rate and performance with methods that require an expensive generic model pre-training.

## 2 BACKGROUND

### 2.1 MODEL COMPRESSION WITH LOW-RANK APPROXIMATION

Given a matrix $W \in \mathbb{R}^{N \times M}$, the low-rank approximation is achieved via singular value decomposition (SVD):

$$W \approx USV^T, \tag{1}$$

where $U \in \mathbb{R}^{N \times r}$, $V \in \mathbb{R}^{M \times r}$, and $k$ is the rank of matrix $W$. $S$ is a diagonal matrix of non-zero singular values $diag(\sigma_1, , ..., \sigma_r)$, where $\sigma_1 \geq \sigma_2 \geq \cdots \sigma_r \geq \cdots \sigma_k > 0$. The low-rank approximation with targeted rank $r$ is obtained by setting zeros to $\sigma_{r+1}, ..., \sigma_k$.

Given input data $X \in \mathbb{R}^{1 \times N}$, a linear layer in neural networks is represented below with the weight matrix $W \in \mathbb{R}^{N \times M}$ and bias $b \in \mathbb{R}^{1 \times M}$:

$$Z = XW + b \approx (XUS)V^T + b. \tag{2}$$

Factorizing W with Equation (1) leads to Equation (2), which can be implemented with two smaller linear layers: 1) The first layer has $Nr$ parameters without bias. Its weight matrix is $US$. 2) The second layer has $Mr$ parameters plus bias. Its weight matrix and bias are $V$ and $b$, correspondingly. The total number of parameters for approximating $W$ is $Nr + Mr$. In the case of full rank matrix and $M = N$, the model size is reduced when $r < 0.5N$. For example, if we set $r$ to reserve the largest 30% singular values, the method will reduce about 40% of the parameters from $W$. In general, the reduced size will be $NM - (Nr + Mr)$.

Low rank approximation in neural networks has been extensively studied (Jaderberg et al., 2014; Zhang et al., 2015; Denton et al., 2014). In more recent works, SVD is often applied to compress the word embedding layer (Chen et al., 2018a; Acharya et al., 2019). Noach & Goldberg (2020)

applies SVD to the transformer layers, but it does not investigate why SVD gives a very poor result without fine-tuning. Our work explores this issue and provides a weighted version to address it.

## 2.2 FISHER INFORMATION

The Fisher information measures the amount of information that an observable dataset $D$ carries about a model parameter $w$. The computation of its exact form is generally intractable since it requires marginalizing over the space of $D$, which includes data and its labels. Therefore, most of the previous works estimate its empirical Fisher information:

$$I_w = E\left[\left(\frac{\partial}{\partial w}\log p(D|w)\right)^2\right] \approx \frac{1}{|D|}\sum_{i=1}^{|D|}\left(\frac{\partial}{\partial w}\mathcal{L}(d_i;w)\right)^2 = \hat{I}_w. \tag{3}$$

The estimated information $\hat{I}_w$ accumulates the squared gradients over the training data $d_i \in \mathcal{D}$, where $\mathcal{L}$ is the target task objective (*e.g.*, cross-entropy for a classification task, or mean squared error for a regression task). This approximation provides a straight intuition: the parameters that change the task objective with a large absolute gradient are important to the target task; therefore, those parameters should be reconstructed better than others in the compression process.

The above estimation of Fisher information computes only the first-order derivatives and has been shown to measure the importance of parameters effectively. Kirkpatrick et al. (2017) and Hua et al. (2021) use it to avoid the model catastrophic forgetting in a continual learning scenario. Liu et al. (2021) and Molchanov et al. (2019) use it or a similar variant to help the structured model pruning. However, no previous work has explored its potential in assisting SVD for model compression.

## 3 MODEL COMPRESSION WITH SVD MAY LOSE PERFORMANCE QUICKLY

The singular values in $S$ implicitly give an importance score for a group of parameters. Since the small singular values will be truncated first, those parameters affected by the truncation are expected to be not important for the task performance. We verify the above assumption with a brute force attack: truncate one singular value at a time, then reconstruct the matrix, put it into a model, evaluate and get its performance. Ideally, we hope to see less performance drop when we truncate the smaller singular values. This process can be written as having the reconstructed model weights $\bar{W}_i$ with the $i$-th singular value be truncated:

$$\bar{W}_i = u_1\sigma_1 v_1^T + ... + u_{i-1}\sigma_{i-1}v_{i-1}^T + u_{i+1}\sigma_{i+1}v_{i+1}^T + ... + u_k\sigma_k v_k^T, \tag{4}$$

where $u_i$ and $v_i$ are the $i$-th column in $U$ and $V$, correspondingly.

Applying this brute force attack to test a deep neural network is not straightforward since a model can have hundreds of linear layers. Therefore, we truncate a group of singular values together instead of only one. Specifically, we split the singular values of a layer into 10 groups sorted by their values. The 1st group has the top 10% singular values, while the 10th group contains the smallest 10%

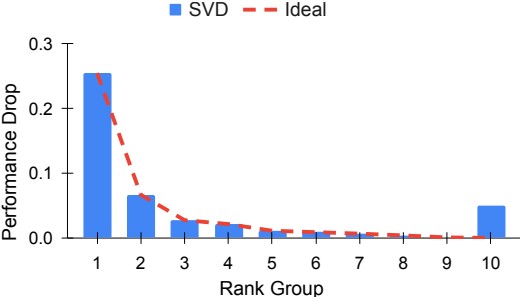

Figure 1: The grouped truncation and its performance. The truncation of the 10th group, which has the smallest singular values resulting from SVD, is expected to have a minor performance impact (*i.e.*, follow the ideal trend of red dashed line), but this may not be true in actual cases (blue bar).

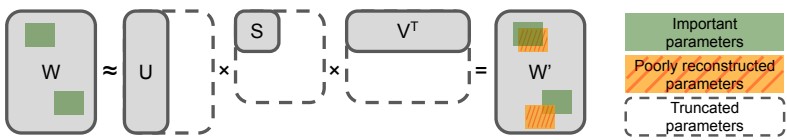

Figure 2: The dilemma of vanilla SVD. Some parameters (the overlap of meshed orange and green) that significantly impact the task performance may not be reconstructed well by SVD because their associated singular values are small and truncated.

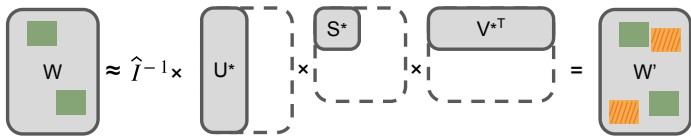

Figure 3: The schematic effect of our Fisher-Weighted SVD (FWSVD). $\hat{I}$ is a diagonal matrix containing estimated Fisher information of parameters. By involving Fisher information to weigh the importance, our method reduces the overlap between meshed orange and green, making less performance drop after truncation.

values. When we truncate a specific group, *e.g.*, 5th group, the 5th group of all the layers in a model are truncated together. In other words, we observe the summed impact in a rank group. This results in a smoothed trend for the observation.

Figure 1 plots the result of truncating the 10 groups separately in a standard 12-layer BERT model (Devlin et al., 2018) trained for STS-B task (Cer et al., 2017). The red dashed line shows an ideal trend which has a smaller performance drop with the tail groups. The blue bars show the actual performance drop. The 10th group surprisingly caused a performance drop as large as the 2nd group. This means the parameters associated with the 10th group are as important as the 2nd group. However, the magnitude of singular value does not reflect this importance, causing a model to lose its performance quickly even when truncating only a small portion.

## 4 FISHER-WEIGHTED LOW-RANK APPROXIMATION

The issue in Section 3 has an intuitive cause: the optimization objective of SVD does not consider each parameter's impact on the task performance. This issue is illustrated in Figure 2, and we address it by introducing the Fisher information into SVD's optimization objective, described as below.

In the generic low-rank approximation, its objective minimizes $||W - AB||_2$. SVD can solve this problem efficiently by having $A = US$ and $B = V^T$. Since we can obtain the importance of each element $W_{ij}$ in $W$, we weigh the individual reconstruction error by multiplying with the estimated Fisher information $\hat{I}_{W_{ij}}$:

$$\min_{A,B} \sum_{i,j} \hat{I}_{W_{ij}} (W_{ij} - (AB)_{ij})^2. \tag{5}$$

In general, weighted SVD does not have a closed-form solution (Srebro & Jaakkola, 2003) when each element has its weight. To make our method easy to deploy and analyze, we propose a simplification by making the same row of the $W$ matrix to share the same importance. The importance for the row $i$ is defined to be the summation of the row, *i.e.*, $\hat{I}_{W_i} = \sum_j \hat{I}_{W_{ij}}$.

Define the diagonal matrix $\hat{I} = diag(\sqrt{\hat{I}_{W_1}}, ..., \sqrt{\hat{I}_{W_N}})$, then the optimization problem of Equation (5) can be written as:

$$\min_{A,B} ||\hat{I}W - \hat{I}AB||_2. \tag{6}$$

Equation 6 can be solved by the standard SVD on $\hat{I}W$. We use the notation $svd(\hat{I}W) = (U^*, S^*, V^*)$, then the solution of Equation (6) will be $A = \hat{I}^{-1}U^*S^*$, and $B = V^{*T}$. In other words, the solution is the result of removing the information $\hat{I}$ from the factorized matrices. Figure 3 illustrates this process and its schematic effect of reducing the overlap between important parameters and poorly reconstructed parameters. We will measure this overlap with the performance drop analysis of Section 3. Lastly, to compress $W$, we will have $A = \hat{I}^{-1}U_r^*S_r^*$, and $B = V_r^{*T}$, where $r$ denotes the truncated $U^*$, $S^*$, and $V^*$ with reserving only $r$ ranks.

We call the above method FWSVD in this paper. One thing to highlight is that since we share the same optimization process with the standard SVD, any advantage we observed will be the result of a direct contribution from the $\hat{I}$ in Equation (6).

## 5 EXPERIMENTS

### 5.1 THE PATHS TO A COMPRESSED LANGUAGE MODEL

This section describes how we obtain a compressed model under the popular pre-training schemes of language models. Figure 4 illustrates three paths that we examined for creating compressed language models. All the paths start from retraining a large transformer-based model pre-trained with a large language corpus in a self-supervised way, called the generic pre-training ($L \rightarrow L^g$).

The path-1 ($S \rightarrow S^g \rightarrow S^t$) is a popular scheme that creates a small model first, then performs the generic distillation for the small model to learn the knowledge of the large model. The resulting small generic model $S^g$ will be fine-tuned with the target task dataset to obtain the task-specific model $S^t$. The representative works of path-1 include DistilBERT (Sanh et al., 2019), TinyBERT (Jiao et al., 2019), MobileBERT (Sun et al., 2020), and MiniLM v1/v2 (Wang et al., 2020b;a). Some previous methods may include task-specific distillation ($L^t \rightarrow S^t$) and data augmentation (Jiao et al., 2019), but we exclude those from the scheme (and all the experiments in this paper) to make a fair and clean comparison across methods. The task-specific distillation and data augmentation are orthogonal to all the methods and can be jointly applied with low-rank factorization to make further improvements.

The path-2 ($L^g \rightarrow L^t \rightarrow L^{tf}$) avoids the costly generic pre-training, directly compresses the task-specific model with factorization and task-specific fine-tuning (optional). Our analysis for the mismatched objectives phenomenon is based on this path. We also compare the models from path-1 and path-2, showing that path-2 can generate a model with a comparable performance under the same compression rate. Although path-2 requires much less training than path-1 (no generic pre-training for the compressed model).

The path-3 ($S^t \rightarrow S^{tf}$) is a challenging setting that aims to compress an already compact model. This setting examines whether FWSVD can further improve the compression rate on models obtained by path-1. Our experiments show the answer is yes.

With the three compression paths, we make four examinations as follows. Section 5.3: the comparison of path-1 versus path-2; Section 5.4: the compression of an already compact model (path-3); Section 5.5: the detailed comparison between FWSVD and vanilla SVD; Section 5.5.1: the empirical evidence for the schematic effects illustrated in Figures 2 and 3.

### 5.2 EXPERIMENT SETUP

#### 5.2.1 LANGUAGE TASKS AND DATASETS

We evaluate the methods of all three paths in Figure 4 on the General Language Understanding Evaluation (GLUE) benchmark (Wang et al., 2019) and a token classification task. We include 2 single sentence tasks: CoLA (Warstadt et al., 2018) measured in Matthew's correlation, SST2 (Socher et al., 2013) measured in classification accuracy; 3 sentence similarity tasks: MRPC (Dolan et al., 2005) measured in F-1 score, STS-B (Cer et al., 2017) measured in Pearson-Spearman correlation, QQP (Chen et al., 2018b) measured in F-1 score; and 3 natural language inference tasks: MNLI (Williams et al., 2018) measured in classification accuracy with the average of the matched and mismatched subsets, QNLI (Rajpurkar et al., 2016) measured in accuracy. The token classification task

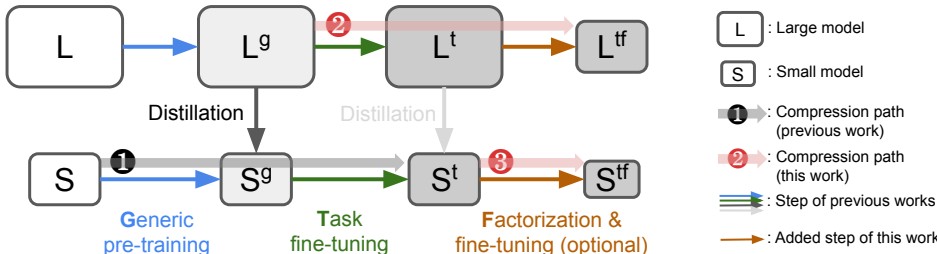

Figure 4: The three paths to create compressed language models are examined in this paper. $L/S$ denote the initial models, $L^g/S^g$ are models after generic pre-training, $L^t/S^t$ correspond to task-specific models, and $L^{tf}/S^{tf}$ are factorized task-specific models. Detailed elaborations are in Sections 5.1 nad 5.2.2

Table 1: Results of CoNLL and GLUE benchmark. G-Avg means the average of GLUE tasks, A-Avg denotes the average of all tasks, including CoNLL. Our FWSVD+fine-tuning is the best performer in terms of both average scores, without the expensive generic pre-training required by path-1 models (e.g., DistillBERT costs 720 V100 GPU hours for training).

| | Model | #Param | CoNLL | CoLA | MNLI | MRPC | QNLI | QQP | SST-2 | STS-B | G-Avg | A-Avg |
|---|---|---|---|---|---|---|---|---|---|---|---|---|
| Original | $\text{BERT}_{base}$ | 109.5M | 94.1 | 56.2 | 84.7 | 87.4 | 91.3 | 87.8 | 93.0 | 88.5 | 84.1 | 85.4 |
| Path-1 | DistilBERT | 67.0M | 93.2 | 49.8 | 82.2 | 88.7 | 89.3 | 86.7 | 90.4 | 86.1 | 81.9 | 83.3 |
| | MiniLMv2 | 67.0M | 92.2 | 43.3 | 84.0 | 89.1 | 90.6 | 86.7 | 91.4 | 88.1 | 81.9 | 83.2 |
| Path-2 | BERT-PKD | 67.0M | − | 45.5 | 81.3 | 85.7 | 88.4 | 88.4 | 91.3 | 86.2 | 81.0 | − |
| | BERT+SVD | 66.5M | 12.0 | 2.7 | 35.6 | 61.4 | 37.2 | 60.0 | 76.7 | 26.8 | 42.9 | 39.0 |
| | +fine-tuning | 66.5M | 92.4 | 40.5 | 82.8 | 84.1 | 89.6 | 87.3 | 90.9 | 85.7 | 80.1 | 81.6 |
| | BERT+FWSVD | 66.5M | 49.6 | 13.5 | 52.8 | 81.2 | 52.2 | 65.7 | 82.1 | 68.6 | 59.4 | 58.2 |
| | +fine-tuning | 66.5M | 93.2 | 49.4 | 83.0 | 88.0 | 89.5 | 87.6 | 91.2 | 87.0 | **82.2** | **83.6** |

we used is the named entity recognition (NER) on the CoNLL-2003 dataset (Sang & De Meulder, 2003). In summary, our evaluation includes 8 different natural language tasks.

### 5.2.2 IMPLEMENTATION DETAILS AND THE BASELINE MODELS

First of all, we use the same training configuration for all the experiments in this paper and avoid any hyperparameter screening to ensure a fair comparison.

For the SOTA models on path-1 (MiniLMv2 and DistilBERT), we use the pre-trained generic compact models ($S^g$) provided by the original authors as the starting point, then directly fine-tune them with 3 epochs on the target task training data. The fine-tuning is optimized by Adam with learning rate of $2 \times 10^{-5}$ and batch size of 32 on one GPU.

For the methods on path-2 (FWSVD and SVD), we start from the pre-trained generic large model ($L^g$), which is the standard 12-layer BERT model (Devlin et al., 2018). Then we fine-tune it with the training setting exactly the same as we used for the path-1 models to get the large task-specific models ($L^t$). The last step is applying the low-rank factorization (SVD or FWSVD) followed by another fine-tuning with the same training setting described above. The performance with and without fine-tuning will be both reported. We also note that we compress only the linear layers in the transformer blocks by reserving only 33% of the ranks in this work. The setup intentionally makes a fair comparison to the path-1 methods. In other words, we do not compress the non-transformer modules such as the token embedding. Previous works (Chen et al., 2018a) have shown significant success in using low-rank factorization to compress the embedding layer, which occupies 23.4M (21.3%) parameters in the standard BERT model. Therefore, the results we reported for the path-2 methods still have room for improvement by applying our method to non-transformer modules. Lastly, we add BERT-PKD (Sun et al., 2019) based on its reproduced results (Chen et al., 2018a) for comparison. BERT-PKD uses knowledge distillation instead of factorization in the path-2 process.

Table 2: Results of compressing an already compact model. The original task-specific models are directly downloaded from Huggingface pretrained models. Our FWSVD successfully reduces more parameters from all the compact models, while achieving the same level of accuracy. (ft: fine-tuning)

| Original Compact Model ($S^t$) | | | Path-3 Compression ($S^t \to S^{tf}$) | | | | |
|---|---|---|---|---|---|---|---|
| Model-Task | #Param. | Perf. | #Param. | SVD | SVD+ft. | FWSVD | FWSVD+ft. |
| TinyBERT-STSB | 14.4M (7.8x) | 87.5 | 11.8M (-18%) | 73.8 | 86.1 | 84.9 | **88.0** |
| MiniLM-CoNLL | 22.7M (4.8x) | 88.5 | 18.4M (-19%) | 12.5 | 88.0 | 70.1 | **88.6** |
| MobileBERT-MNLI | 24.6M (4.4x) | **83.6** | 22.5M (-9%) | 36.4 | 81.9 | 51.1 | 82.5 |
| DistillBERT-MRPC | 66.9M (1.6x) | 88.7 | 46.7M (-30%) | 0.0 | 83.4 | 67.9 | **89.0** |

Table 3: Results of compressing an already compact model. This table compresses ALBERT (Lan et al., 2019), which uses the parameter-sharing strategy to create the compact model. FWSVD preserves the performance significantly better than SVD in all 8 tasks, indicating its excellent compatibility in combining the parameter-sharing strategy. This experiment examines the path-3 process.

| Model | #Param | CoNLL | CoLA | MNLI | MRPC | QNLI | QQP | SST-2 | STS-B | G-Avg | A-Avg |
|---|---|---|---|---|---|---|---|---|---|---|---|
| BERT$_{base}$ | 109.5M | 94.1 | 56.2 | 84.7 | 87.4 | 91.3 | 87.8 | 93.0 | 88.5 | 84.1 | 85.4 |
| ALBERT$_{large}$ | 17.7M | 93.5 | 50.9 | 84.3 | 89.9 | 91.7 | 86.7 | 90.7 | 90.1 | 83.5 | 84.7 |
| w SVD | 15.2M (-14%) | 0.3 | 0.0 | 41.1 | 0.0 | 54.2 | 5.4 | 70.2 | 9.6 | 25.8 | 22.6 |
| w SVD+ft. | 15.2M (-14%) | 92.2 | 46.4 | 83.4 | 81.8 | 49.5 | 86.9 | 89.8 | 86.8 | 74.9 | 77.1 |
| w FWSVD | 15.2M (-14%) | 22.8 | 0.0 | 65.2 | 50.0 | 78.2 | 72.6 | 81.4 | 76.4 | 60.5 | 55.8 |
| w FWSVD+ft. | 15.2M (-14%) | 93.0 | 50.6 | 83.3 | 90.4 | 90.6 | 87.0 | 90.6 | 89.0 | **83.1** | **84.3** |
| ALBERT$_{base}$ | 11.7M | 92.1 | 43.0 | 82.3 | 88.6 | 90.6 | 86.6 | 89.7 | 89.1 | 81.4 | 82.7 |
| w SVD | 9.6M (-18%) | 3.5 | 0.0 | 32.0 | 0.0 | 55.1 | 53.4 | 52.4 | 9.6 | 28.9 | 25.7 |
| w SVD+ft. | 9.6M (-18%) | 89.8 | 28.8 | 81.3 | 81.2 | 88.3 | 85.5 | 88.2 | 75.0 | 75.5 | 77.3 |
| w FWSVD | 9.6M (-18%) | 16.9 | 6.9 | 55.6 | 47.7 | 69.1 | 54.7 | 72.9 | 54.0 | 51.6 | 47.2 |
| w FWSVD+ft. | 9.6M (-18%) | 91.2 | 42.2 | 81.8 | 86.9 | 88.9 | 86.2 | 88.7 | 87.0 | **80.2** | **81.6** |

For the path-3 experiments, we use the pre-trained task-specific *compact* models ($S^t$) as the starting point. These pre-trained models have a much smaller size (TinyBERT-STSB, MiniLM-CoNLL, MobileBERT-MNLI) or a better performance (DistilBERT-MRPC) than the models we used in the path-1 and path-2, indicating they may contain denser knowledge in their compact models. Therefore, compressing these models introduces a significant challenge. In order to have better coverage for all tasks, we additionally use ALBERT$_{large}$ and ALBERT$_{base}$ (Lan et al., 2019) as the already compact models to generate all 8 task-specific models ($S^g \to S^t$). Then follow path-3 to compress the compact models. All the training involved here has the same setting as described in path-1.

Lastly, our implementation and experiments are built on top of the popular HuggingFace Transformers library (Wolf et al., 2020). All other unspecified training settings use the default configuration of the library. Since no hyperparameter tuning is involved in our experiments, we directly report the results on the dev set of all the datasets, making the numbers convenient to compare and verify.

## 5.3    PATH-1 VERSUS PATH-2

Table 1 reports the results of the GLUE benchmark and a NER task. Our FWSVD with fine-tuning achieves an average score of $83.6$, beating all other path-1 and path-2 methods. This is a non-trivial accomplishment since FWSVD with fine-tuning does not need the expensive generic pre-training. Furthermore, FWSVD has consistent performance retention for all the tasks; it contrasts the path-1 methods, which may have a more considerable variance. For example, DistilBERT is good at CoLA but poor at STS-B; oppositely, MiniLMv2 is a strong performer at STS-B but is weak with CoLA. In contrast, FWSVD+fine-tuning does not show an obvious shortcoming.

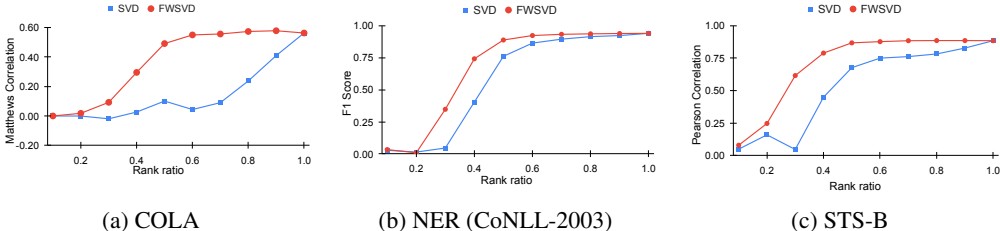

(a) COLA             (b) NER (CoNLL-2003)           (c) STS-B

Figure 5: FWSVD versus SVD by varying the ratio of reserved ranks. The model with a rank ratio $1.0$ indicates the full-rank reconstruction with the same accuracy as the original model (*i.e.*, the $L^t$ in Figure 4). Note that all the models here do not have fine-tuning after factorization.

## 5.4 COMPRESSING AN ALREADY COMPACT MODEL

The setting of path-3 targets to further compress the lightweight models. This is challenging as the compact models are already $1.6x \sim 7.8x$ smaller than the original BERT. The results in Table 2 demonstrate the effectiveness of FWSVD on further reducing the number of parameters. The original SVD is almost useless without fine-tuning, while our FWSVD can still retain a significant part of the performance. For example, SVD ends with a zero accuracy when compressing DistillBERT, while our FWSVD keeps a score of $67.9$ under the same setting. When combined with fine-tuning, FWSVD can cut off 30% redundancy for DistillBERT. Even for the highly compact model TinyBERT (only $14.4M$ parameters), FWSVD+fine-tuning still successfully reduces 18% of the parameters without any performance loss. More interestingly, the TinyBERT, MiniLM, and DistillBERT-MRPC compressed by FWSVD+fine-tuning exceed the original performance slightly. The result suggests FWSVD+fine-tuning might introduce a small regularization effect to improve the model's generalizability.

Lastly, Table 3 examines the compatibility between SVD/FWSVD and the parameter-sharing strategy of the ALBERT model. The average score of ALBERT-large is 84.7%. The performance of FWSVD (84.3%) is far better than that of SVD (77.1%) when both reducing 14% parameters, suggesting FWSVD is more robust than SVD in combining the parameter-sharing strategy.

## 5.5 FWSVD VERSUS SVD

In Table 1, FWSVD consistently produces much better results than SVD on all tasks. On average, FWSVD without fine-tuning obtains an absolute improvement of 17.5% over SVD. To highlight, FWSVD without fine-tuning can maintain a significant portion of performance in challenging tasks such as CoNLL and STS-B, where SVD completely fails. With fine-tuning, FWSVD provides better initialization for fine-tuning and consistently achieves a better or comparable performance.

Figure 5 plots the performance trend with respect to the change of targeted rank ratio, where the full-rank reconstruction corresponds to the results at rank ratio $1.0$. These results demonstrate the apparent advantage of FWSVD over standard SVD. First, at each rank ratio, FWSVD shows significant improvements over SVD. Second, the performance of FWSVD keeps growing with the increase of rank ratio, while SVD shows fluctuations in its trend. Specifically, two tasks (COLA and STS-B) in Figure 5 show that SVD has abrupt performance drops at some points. On the STS-B task, the performance of SVD at rank ratio $0.3$ is significantly lower than having a smaller rank ratio of $0.2$. In contrast, FWSVD shows a much stable trend of increasing performance along with the rank ratio.

### 5.5.1 REVISIT THE BRUTE FORCE ATTACK

This section applies the same analysis of Section 3, but adds FWSVD to see if it matches the task's objective better. In Figure 6a, the red bars are significantly lower than the blue bars, especially for the tail groups, which will be truncated first. We specifically highlight group-10 in 6a, which has the smallest 10% singular values. The height of the blue bar is equivalent to the size of the overlapped (green and meshed orange) region in Figures 2. Similarly, its red bar (close to zero) is equivalent to the overlapped region in Figure 3. In other words, the illustrations of Figures 2 and 3 are strongly supported by the results here. Although FWSVD shows a smaller performance drop

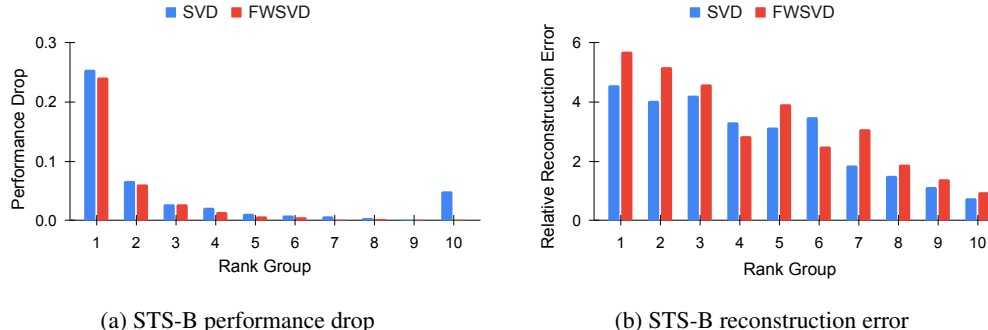

(a) STS-B performance drop  (b) STS-B reconstruction error

Figure 6: Results of grouped rank truncation on STS-B task. In (a), FWSVD shows a consistent trend of having less performance drop with the small singular value groups (group 10 has the smallest singular values), mitigating the issue of Figure 1. In (b), FWSVD results in a larger reconstruction error with almost all truncated groups, although FWSVD retains the model accuracy better than SVD.

shown by Figure 6a, it has a more significant reconstruction error than SVD in many cases (see Figure 6b), especially with the rank groups that will be truncated first (*e.g.*, groups 5 to 10). In other words, FWSVD's objective (Equation 6) aligns with the task objective better by sacrificing the reconstruction error.

## 6    LIMITATION AND FUTURE WORK

FWSVD has two limitations. First, FWSVD relies on a given task objective and a target task training dataset to compute the importance matrix; thus, it is more proper to compress a task-specific model (*e.g.*, $L^t$ or $S^t$) than the pre-trained generic model (*e.g.*, $L^g$). In contrast, the vanilla SVD can apply to any case. In other words, FWSVD trades the method's applicability for the target task performance. Second, FWSVD only uses a simplified importance matrix that gives the same importance for the parameters on the same row of matrix $W$. Although this strategy is simple and effective, it does not fully utilize the Fisher information. Therefore, a future improvement can be made by directly seeking an element-wise factorization solution for Equation (5).

## 7    CONCLUSION

In this work, we investigate why using standard low-rank factorization (SVD) to compress the model may quickly lose most of its performance, pointing out the issue of the mismatched optimization objectives between the low-rank approximation and the target task. We provide empirical evidence and observations for the issue, and propose a new strategy, FWSVD, to alleviate it. Our FWSVD uses the estimated Fisher information to weigh the importance of parameters for the factorization and achieve significant success in compressing an already compact model. Furthermore, FWSVD reuses the existing SVD solver and can still implement its factorized matrices with linear layers; therefore, it is simple to implement and deploy. We believe FWSVD could be one of the most easy-to-use methods with good performance for language model compression.

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

# A Supplementary

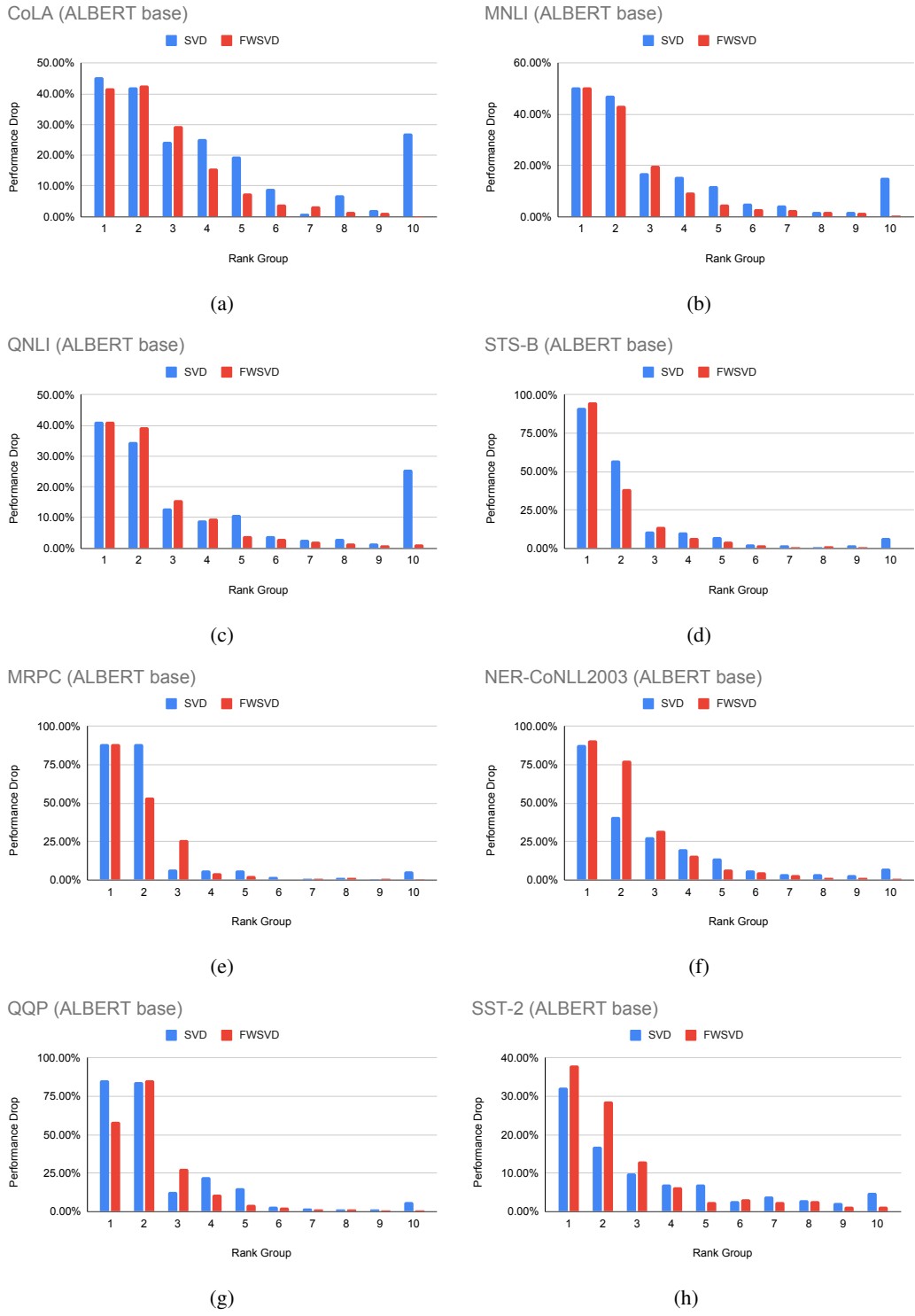

Figure 7: The grouped rank truncation experiment. The experiments are the same with Figure 6a, but we use ALBERT$_{base}$ (11.7M parameters) model for this figure.

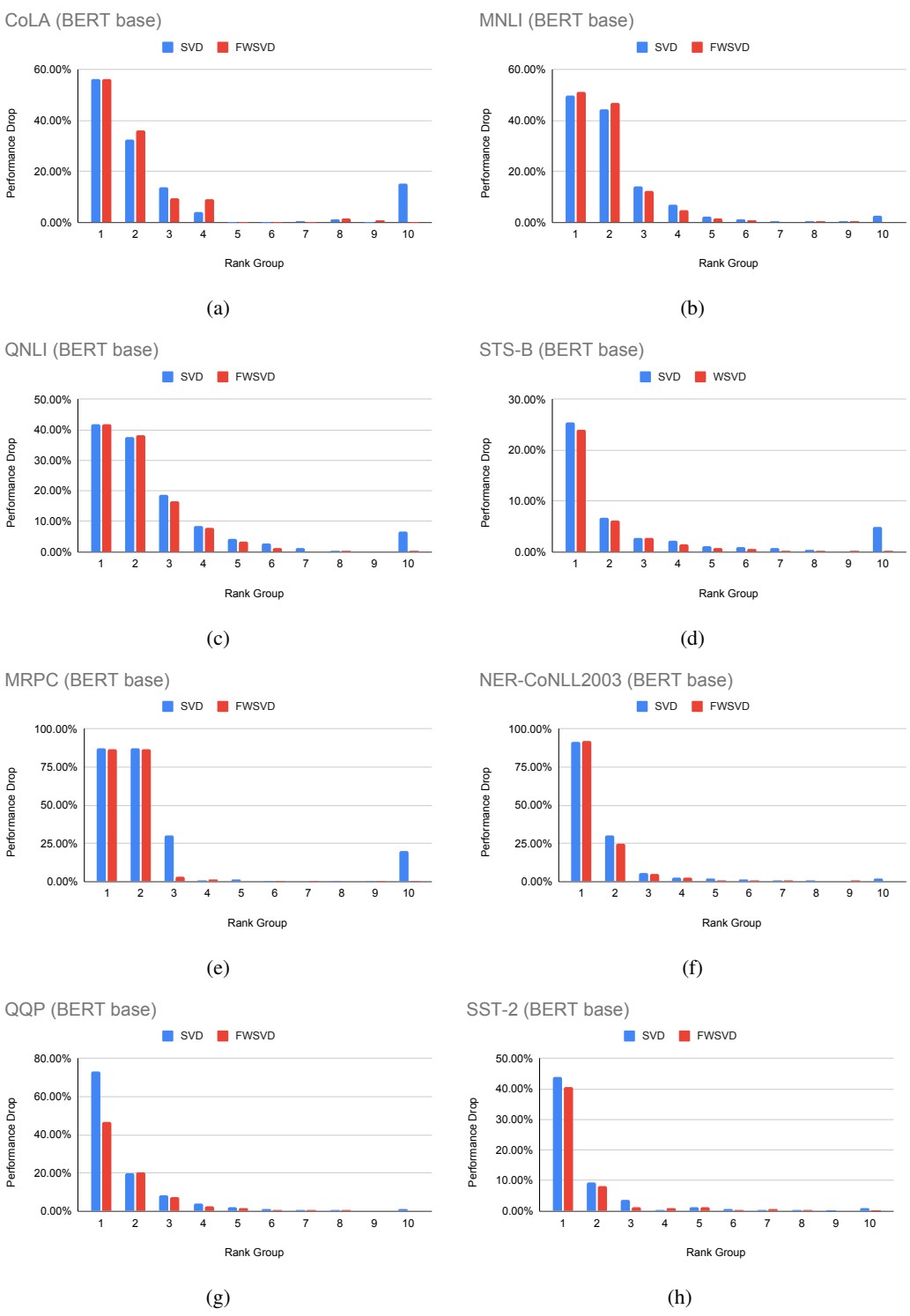

Figure 8: The grouped rank truncation experiment. The experiments are the same with Figure 6a, but this figure includes all 8 language tasks with BERT$_{base}$ (109.5M parameters) model. FWSVD has a smaller performance drop with those groups truncated first (*e.g.*, group 5 to 10) in all the cases. SVD usually shows a significant drop with group 10, which has the smallest singular values and is truncated first. FWSVD has no such issue in all cases.

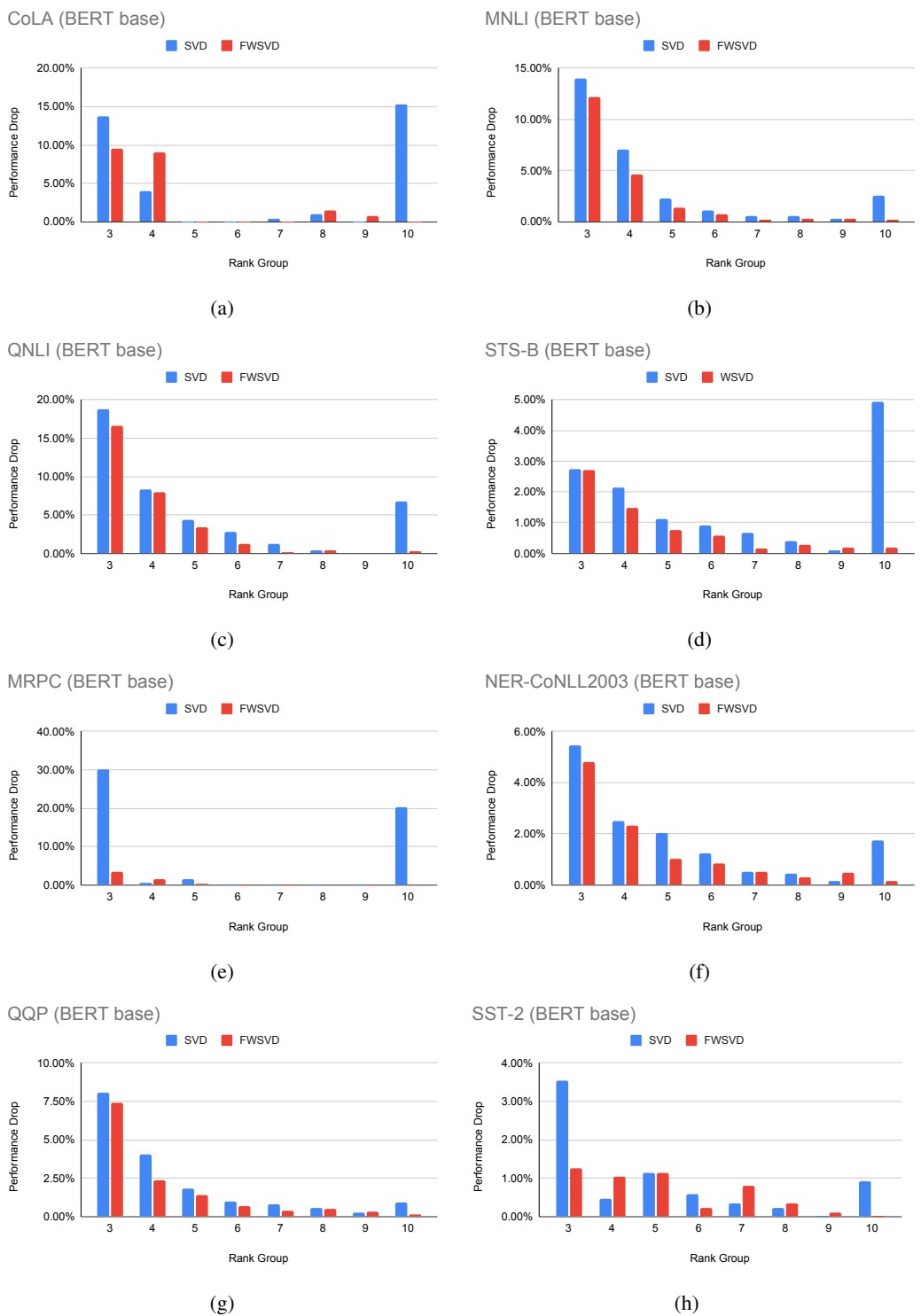

Figure 9: This figure shows only groups 3 to 10 of Figure 8 to better visualize the groups of a smaller performance drop.

Table 4: The raw values for Figure 6a. We additionally include the averaged singular values for each truncated group. The singular values from FWSVD are multiplied with Fisher information; thus their scales are different from SVD.

| Truncated group | 1 | 2 | 3 | 4 | 5 | 6 | 7 | 8 | 9 | 10 |
|---|---|---|---|---|---|---|---|---|---|---|
| SVD performance drop | 25.4% | 6.7% | 2.8% | 2.2% | 1.1% | 0.9% | 0.7% | 0.4% | 0.1% | 4.9% |
| FWSVD performance drop | 24.1% | 6.1% | 2.7% | 1.5% | 0.8% | 0.6% | 0.2% | 0.3% | 0.2% | 0.2% |
| SVD average singular value | 2.674 | 1.933 | 1.622 | 1.381 | 1.176 | 0.994 | 0.828 | 0.671 | 0.519 | 0.353 |
| FWSVD average singular value | 1093 | 631 | 510 | 424 | 355 | 298 | 247 | 201 | 157 | 110 |

