# OpenReview forum: "Language model compression with weighted low-rank factorization"
_ICLR.cc/2022/Conference — ICLR 2022 Poster_

### Official Review · Reviewer_d1T4 · 2021-11-01

**Correctness:** 4
**Technical Novelty And Significance:** 3
**Empirical Novelty And Significance:** 3
**Recommendation:** 6
**Confidence:** 4

**Main Review:**

Strengths:
1. Solid experiments demonstrating the proposed method outperforms other existing approaches.
2. Novel weighting scheme for SVD for low-rank weight compression (though should double check this more thoroughly).



Some comments:

1. I would suggest using the phrase "weighted SVD" early on in the introduction (e.g., exactly when you introduce your new method).
2. It might be good to address the existing literature on compressing trained neural networks which also goes beyond simply trying to minimize the Frobenius norm of the difference between the weights. For example, the paper https://arxiv.org/abs/1505.06798 also goes beyond simply considering the reconstruction objective on the weights, and includes the nonlinearity as well in the reconstruction objective. Your work obviously is different enough to stand on its own, but it might be good to make a mention of this work and others (e.g. do more of a lit search / related work on low rank compression).
3. It might be good to find the best hyperparameters for each setting to truly do a fair comparison (avoiding hyperparameter tuning isn't really a fair comparison, IMO -- but this depends on how much compute you have available).
4. You forgot to bold the best performer in line 3 of Table 2 (in this case, the original compact model).


**Summary Of The Paper:**

This paper studies the compression of the weights of a neural network via a modification of the standard paradigm of using low rank approximations via SVD on the weight matrices. Instead, they propose a weighted SVD objective where the weights are computed based on the Fisher information of the (trained) parameter with respect to the dataset. They simplify the method so it is more computationally tractable by having the same row of each weight matrix share the same weight. This translates to solving SVD on a modified reconstruction objective (multiply by a diagonal). Here the Fisher information for parameter w is approximated by the average squared l2 norm of the gradient with respect to w over the empirical data distribution.

They then conduct several experiments with this approach in the problem domain of compressing language models. They evaluate their method on compressing task specific models as well as the problem of compressing an already compressed model, and demonstrate their approach is generally better.



**Summary Of The Review:**

The paper introduces a new approach to weighting the low-rank compression of neural nets, and empirically demonstrates that it outperforms other methods. The approach isn't too surprising, but it does appear to be novel (though I would recommend doing a more thorough lit review and make the related work section larger, since I feel certain that there are many more related papers doing similar things (I mention one in the main review). I think the experiments are overall good and demonstrate the point nicely, which is essentially that the important thing in compressing networks is not the parameter compression -- but rather, the compression of the function specifically on the relevant distribution. Overall I vote to accept.

UPDATE:
I read the comments, my opinion is unchanged.

---

> ### Author Response · Authors · 2021-11-23
> **Response**
>
> We greatly appreciate Reviewer d1T4's suggestions for improvement and have incorporated these into our submission.
>
> **1. Using the phrase "weighted SVD" early on in the introduction**
>
> Good suggestion. We have revised it.
>
> **2. Include other method that also goes beyond simply trying to minimize the Frobenius norm.**
>
> Thanks for the suggested reference. We have added it as well as its related literature.
>
> **3. It might be good to find the best hyperparameters for each setting to truly do a fair comparison (avoiding hyperparameter tuning isn't really a fair comparison, IMO -- but this depends on how much compute you have available)**
>
> Thanks for the suggestion, and you are right about the limited computational budget. We believe our method can benefit many developers who do not have luxury computation resources.
>
> **4. Bold the best performer in line 3 of Table 2.**
>
> Thank you so much for catching the mistake, and we have fixed it.

---

### Official Review · Reviewer_Kuwu · 2021-11-02

**Correctness:** 3
**Technical Novelty And Significance:** 2
**Empirical Novelty And Significance:** Not applicable
**Recommendation:** 6
**Confidence:** 4

**Main Review:**


-) Written English needs improvement.

-) The idea is interesting and the authors explain the main intuition adequately clear.
The authors explain the advantages and limitations properly.

-) How easy/fast is to generate the fine-tuned model? In other words, what is the overall
computational cost to create a fine-tuned model and apply FWSVD versus generic training
and SVD?

-) Unfortunately it is not clear to me whether the proposed method comes with any guarantees.
I understand that minimizing the Euclidean norm of the approximation of W does not necessarily
imply lower classification (or other tasks) error, however I am not sure the proposed method does
either. Is that so? For example, will the new method always be better than SVD? One needs to
account for the cost of pre-processing costs too.

-) In addition to my comment above, Figure 6 seems to complicate the discussion. The legend reads
as "Unlike SVD, the performance drop of our FWSVD perfectly imitates the ideal trending shown in Figure 1.",
however I can see that on the left subfigure "Rank group 8"  leads to a higher performance drop than "Rank group 7" which
is exactly what the authors wanted to avoid in the first place. Furthermore, on the right subfigure, FWSVD
actually introduces higher reconstruction error, which does not even reduce monotonically. I understand that
the goal is to achieve higher accuracy in the classification task and not the matrix reconstruction part,
but this seems to complicate the optimization part of the model, e.g., how do the authors make sure
they do not overfit or spend more time than necessary computing the low-rank approximation?

-) The above experiment is explained better in Section 5.5.1 but the following sentence is confusing
"The results suggest that FWSVD’s objective (Equation 6) aligns with the task objective better by sacrificing the reconstruction error."

-) Section 5.2.2 is rather unclear; can you please provide more details, perhaps in the appendix? Without
proper explanation in this step, the experiments can not be judged fairly.

-) Please make the y-axis in Figure 6 run in logarithmic scale.

-) While generally providing better results than SVD for the experiments shown in this paper, the proposed
method does not offer dramatic improvements in accuracy, nor it is guaranteed to do so. It would be very
helpful if the authors could provide additional experiments about other benefits of the method, if applicable,
in the appendix (e.g., is the new method faster than previous approaches based on SVD etc.)

Minor comments:

-) Equation (1) and related text is confusing. The "\approx" symbol should be "=" since matrix W is of rank 'k'. I suggest the authors to simplify discussion by using 'r' instead of 'k' and simply mention that matrix W is of rank k\geq r.

-) "we propose a simplification by making the same row of the W matrix to share the same importance". Did you mean to say that each weight located in row 'i' will use the average weight of its row?

**Summary Of The Paper:**

The proposed paper focuses on a new technique to compress weight matrices in Machine Learning, e.g., weight matrices of layers in DNN.
One idea is to simply exploit truncated SVD. While this minimizes the Euclidean norm of the error, it doesn't necessarily lead to a lower task error since the truncated part might still have a noticeable effect in the resolution power of the model. The authors suggest to solve an alternative minimization problem which takes empirical Fisher information under consideration. Experiments on a few datasets imply that the proposed method can achieve better results than simply using truncated SVD of the weight matrix.

**Summary Of The Review:**

The proposed idea is interesting and seems to lead to improvements over truncated SVD, however these improvements
are not major. Moreover, the proposed algorithm is less general than truncated SVD.

---

> ### Author Response · Authors · 2021-11-23
> **Response (1/2)**
>
> We greatly appreciate Reviewer Kuwu's comments for improvement. We have incorporated these suggestions and revised our submission.
>
> **1. Written English needs improvement.**
>
> We thank you for the feedback and have revised the sections/sentences mentioned by Reviewer Kuwu.
>
> **2. How easy/fast is to generate the fine-tuned model? In other words, what is the overall computational cost to create a fine-tuned model and apply FWSVD versus generic training and SVD?**
>
> FWSVD has only one overhead beyond a standard SVD method: computing the fisher information $\hat{I_w}$. The computation time for $\hat{I_w}$ is the same as one epoch of normal training. For example, to generate a compressed BERT-base model for STS-B, FWSVD only takes about 1 minute to compute $\hat{I_w}$ and 20 seconds to compute Eq.(6) with a standard SVD solver. The fine-tuning of FWSVD is the same as the fine-tuning of SVD, which is equivalent to the generic training. In other words, FWSVD introduces only minor overhead on top of SVD to generate a compressed model.
>
> **3. Any guarantees of the proposed method? Will the new method always be better than SVD? Is it worth to pay the overhead of FWSVD?**
>
> We have added more results of compressing compact models to Table 3 in the revised submission. Total combinations reach 24 (8 tasks with 3 different models: BERT-base, ALBERT-base, and ALBERT-large). In all cases, FWSVD is significantly better or comparable with SVD. The results suggest that the improvement is generic, and the small computation overhead (1 epoch of training time) is worthy.
>
> **4. Figure 6 seems to complicate the discussion. "Rank group 8" leads to a higher performance drop than "Rank group 7" which is exactly what the authors wanted to avoid in the first place.**
>
> Thanks for the feedback, and we do find the caption of Figure 6 caused confusion. We have revised the caption to improve its clarity. To be more accurate, Figure 6 shows that FWSVD produces the desired trend better than SVD.
>
> **5. How do the authors make sure they do not overfit or spend more time than necessary computing the low-rank approximation.**
>
> FWSVD does the same computation as a standard SVD except that FWSVD makes the input to an SVD solver be weighted by Fisher information $\hat{I_w}$. FWSVD shares the same process with the standard SVD method in all other parts. Besides, there is no hyperparameter to tune in both FWSVD and SVD, thus avoiding unnecessary fitting computation.
>
> **6. The above experiment is explained better in Section 5.5.1 but the following sentence is confusing "The results suggest that FWSVD’s objective (Equation 6) aligns with the task objective better by sacrificing the reconstruction error."**
>
> This sentence summarizes its section and relates to the observation of Section 3. We see the wording may cause some confusion and have revised it in the updated submission. We thank Reviewer Kuwu for raising this concern.
>
> **7. Please make the y-axis in Figure 6 run in logarithmic scale.**
>
> We see Reviewer Kuwu's concern about visualization and provide a zoomed version of Figure 6a in supplementary Figure 9d. We also provide the raw values below and supplementary Table 4.
>
> |(Figure 6a)Truncated group|1|2|3|4|5|6|7|8|9|10|
> |-|-|-|-|-|-|-|-|-|-|-|
> |SVD performance drop|25.4%| 6.7%| 2.8%| 2.2%| 1.1%| 0.9%| 0.7%| 0.4%| 0.1%| 4.9%|
> |FWSVD performance drop|24.1%| 6.1%| 2.7%| 1.5%| 0.8%| 0.6%| 0.2%| 0.3%| 0.2%| 0.2%|
>
> **8. It would be very helpful if the authors could provide additional experiments about other benefits of the method, if applicable, in the appendix (e.g., is the new method faster than previous approaches based on SVD etc.)**
>
> The strength of FWSVD is in compressing an already compact model. We provide additional experiments in compressing a high-performing compact model, ALBERT-large (17.7M). The average score of ALBERT-large is 84.7%. ALBERT+FWSVD achieves 84.3% while reducing additional 14% parameters. The performance is far better than 77.1% of SVD. In our additional tests, SVD needs a larger model (almost equivalent to the original model) to achieve the performance of FWSVD.
>
> |Model|#Param|CoNLL|CoLA|MNLI|MRPC|QNLI|QQP|SST-2|STS-B|A-Avg|
> |-|-|-|-|-|-|-|-|-|-|-|
> |ALBERT$_{large}$|17.7M|93.5|50.9|84.3|89.9|91.7|86.7|90.7|90.1|**84.7**|
> |with SVD|15.2M (-14%)|0.3|0.0|41.1|0.0|54.2|5.4|70.2|9.6|22.6|
> |with SVD+ft.|15.2M (-14%)|92.2|46.4|83.4|81.8|49.5|86.9|89.8|86.8|77.1|
> |with FWSVD|15.2M (-14%)|22.8|0.0|65.2|50.0|78.2|72.6|81.4|76.4|55.8|
> |with FWSVD+ft.|15.2M (-14%)|93.0|50.6|83.3|90.4|90.6|87.0|90.6|89.0|**84.3**|
>
> (continue in next post)

---

> > ### Author Response · Authors · 2021-11-23
> > **Response (2/2)**
> >
> > **9. Section 5.2.2 is rather unclear; can you please provide more details, perhaps in the appendix?**
> >
> > Section 5.2.2 supplements Section 5.1, which summarizes the previous practices of using pre-trained language models. The SVD/FWSVD methods apply to trained tasks specific models ($L^t$ or $S^t$ in Figure 4) and include only two steps: (1) factorize the model weights and (2) fine-tune the factorized model with standard training. We have made minor revise to make the above connections more explicit. We hope this improves clarity.
> >
> > **10. Equation (1) and related text is confusing.**
> >
> > Thanks for pointing out the issue. We have revised it in our updated submission.
> >
> > **11. "we propose a simplification by making the same row of the W matrix to share the same importance". Did you mean to say that each weight located in row 'i' will use the average weight of its row?**
> >
> > The importance for the row $i$ is defined to be the summation of the row (Its formulation locates below Eq.(5)). Using the average weight will have the same result since the inversed Fisher information will remove the scaling factor (in the formulation for $A$ in Section 4).
> >
> > **12. These improvements are not major**
> >
> > Our method targets replacing only SVD. In order to achieve a high compression rate, multiple compression methods are usually combined. For example, parameter-sharing, SVD, and quantization are popular combinations in industrial practice. This work significantly addresses the performance degradation caused by SVD, making the low-rank factorization a more robust strategy to be combined with others.

---

> ### Author Response · Authors · 2021-11-29
> **Followup**
>
> Dear reviewer, do our responses address the concerns? Please feel free to share your thoughts with us. We greatly appreciate your feedback. Thank you very much.

---

### Official Review · Reviewer_jnTC · 2021-11-02

**Correctness:** 4
**Technical Novelty And Significance:** 2
**Empirical Novelty And Significance:** 3
**Recommendation:** 6
**Confidence:** 4

**Main Review:**

The paper is well-written and easy to follow.

Strengths:
+ the approach does not require re-training (doing pre-training again)
+ experimental results on several tasks show only a small drop in model quality
+ the approach can further reduce the size of already-reduced-size models (e.g. TinyBERT)

Weaknesses:
- the reduction in the number of parameters is modest compared to methods that re-train the network in a compact representation, like ALBERT
- Results in Table 2 look incomplete: why is only one task reported for each of the models? Why different task is used for each different model? It would be better to pick e.g. three tasks and use them for all four models.
- the work does not provide much insight into the problem that motivates the approach (e.g. why "group 10" would affect "important parameters" more than "group 9")

Questions:
- how common is the behavior in Fig 1? It would be interesting to see evidence of it on at least one more dataset/task. It might also be useful to use log scale for the axis, currently it is not possible to see if e.g. removing group 9 leads to lower/similar/higher performance drop vs group 8 (i.e., is group 10 special, if yes, why?).
- what is the average/median magnitude of singular values in group 9 vs in group 10?
- have you tried minimizing eq. (5) using SGD? Do you expect substantial gap between (5) and the simplified version in eq. (6)?


---
Post-rebuttal:
Thank you for the clarifications and additional experiments, I've updated my score to 6.

**Summary Of The Paper:**

SVD of matrices in a language model is used to reduce model size. Instead of standard SVD that minimizes the MSE of reconstructing the matrix elements, a modified version that weights elements is used. Specifically, importance of each parameter (matrix element) is estimated based on the magnitude of gradients w.r.t. to that element. To have closed-form solution for weighted SVD, average row weights are used instead of element weights.

**Summary Of The Review:**

The paper provides a simple method to reduce (modestly) the size of an already trained language model, without the need for costly re-training. The evidence in favor of the method is empirical, with little in terms of theoretical or experimental exploration of the motivating phenomenon: the overlap between "important parameters" and "parameters poorly captured by SVD".

---

> ### Author Response · Authors · 2021-11-23
> **Response**
>
> We thank reviewer jnTC's thoughtful feedback. We address the concerns and questions below.
>
> **1. The reduction in the number of parameters is modest compared to ALBERT.**
>
> FWSVD can compress ALBERT to generate a model smaller than ALBERT. For example, we use ALBERT-large (17.7M parameters) to train the 8 tasks, then apply FWSVD to compress these models. FWSVD can reduce additional 14% parameters with the same average performance. The average score of ALBERT is 84.7%, while ALBERT+FWSVD is 84.3%. We include the averaged score (A-Avg) below and have the detailed results in Table 3 of our revised submission.
>
> |-|BERT$_{base}$|ALBERT$_{large}$|with SVD|with FWSVD|with SVD+ft.|with FWSVD+ft.|
> |-|-|-|-|-|-|-|
> |#param(M)|109.5|17.7 (6.2x)|15.2 (7.2x)|15.2 (7.2x)|15.2 (7.2x)|15.2 (7.2x)|
> |A-Avg|85.4|84.7|22.6|55.8|77.1|84.3|
>
> **2. Results in Table 2 look incomplete: why is only one task reported for each of the models? Why different task is used for each different model?**
>
> The models used in Table 2 are directly downloaded from Huggingface model zoo (https://huggingface.co/models). The model zoo does not have all task-specific models. We understand reviewer jnTC's concern and use ALBERT-large and ALBERT-base as the already compact models to generate each task-specific model to extend the experiments of Table 2. We show the results of compressing ALBERT-large below and have the rest in Table 3 of our revised submission.
>
> |Model|#Param|CoNLL|CoLA|MNLI|MRPC|QNLI|QQP|SST-2|STS-B|A-Avg|
> |-|-|-|-|-|-|-|-|-|-|-|
> |ALBERT$_{large}$|17.7M|93.5|50.9|84.3|89.9|91.7|86.7|90.7|90.1|**84.7**|
> |with SVD|15.2M (-14%)|0.3|0.0|41.1|0.0|54.2|5.4|70.2|9.6|22.6|
> |with SVD+ft.|15.2M (-14%)|92.2|46.4|83.4|81.8|49.5|86.9|89.8|86.8|77.1|
> |with FWSVD|15.2M (-14%)|22.8|0.0|65.2|50.0|78.2|72.6|81.4|76.4|55.8|
> |with FWSVD+ft.|15.2M (-14%)|93.0|50.6|83.3|90.4|90.6|87.0|90.6|89.0|**84.3**|
>
> **3. Insight of why "group 10" would affect "important parameters" more than "group 9"?**
>
> This is a great question. The question is equivalent to asking why SVD's singular values can not reflect the importance of parameters. We guess the singular value may reflect the importance in some cases (ex: parameters with a large absolute value) but not the other case.
>
> **4. How common is the behavior in Fig 1?**
>
> Figure 1 is a high-frequency behavior. We have added 16 experiments in supplementary Figures 7 and 8 for the evidence. These plots cover 8 different tasks with 2 different models, BERT-base (109.5M) and ALBERT-base (11.7M). The figures show that SVD causes 13 out of 16 cases a significant performance drop with group 10. FWSVD largely mitigates this behavior in all cases.
>
> **5. What is the average/median magnitude of singular values in group 9 vs in group 10?**
>
> The average magnitude of singular values is 0.519 and 0.353 for groups 9 and 10, respectively. We provide average singlar values for all groups of Figure 1 (and Figure 6a) below and supplementary Table 4.
>
> |Truncated group|1|2|3|4|5|6|7|8|9|10|
> |-|-|-|-|-|-|-|-|-|-|-|
> |SVD performance drop|25.4|6.7|2.8|2.2|1.1|0.9|0.7|0.4|0.1|4.9|
> |SVD average singular value|2.674|1.933|1.622|1.381|1.176|0.994|0.828|0.671|0.519|0.353|
>
> **6. Have you tried minimizing eq. (5) using SGD? Do you expect substantial gap between (5) and the simplified version in eq. (6)?**
>
> Yes, we have tried minimizing Eq.(5) using SGD. Eq.(5) leads to a mildly better result than Eq.(6) before finetuning. However, the performance is the same after fine-tuning. This indicates that Eq.(6) can provide a good-enough model initialization to reach an equivalent local minimum after fine-tuning.
>
> **7. The evidence in favor of the method is empirical**
>
> We add additional empirical evidence in Table 3, Figure 8, and Figure 9 to support our claims. We hope the provided evidence alleviates the concern.

---

### Decision · Program_Chairs · 2022-01-20

**Decision:**

Accept (Poster)

**Comment:**

The paper studies the problem of task-specific model compression obtained from fine-tuning large pre-trained language models. The work follows the line of research in which model size is reduced by decomposing the matrices in the model into smaller factors. Two-step approaches apply SVD and then fine-tuned the model on task specific data. The present work makes the observation that after the first step (the SVD compression) the model can dramatically lose its performance, due to the mismatched optimization objectives between the low-rank approximation and the target task. The work provides evidence backing this claim. The paper proposes to address this problem by weighting the importance of parameters for the factorization according to the Fisher information. Experimental evaluation shows that the proposed method can achieve better results than variants that use truncated SVD of the weight matrices.

The paper is well written and easy to read. The method is simple and effective and can be applied to in a wide range of settings. The authors provided a thorough response which clarified several points. This led Reviewer Kuwu to increase the score to 6.

All three reviewers agree that the main observation in the work is interesting and informative for researchers and practitioners working on the problem.

Reviewer jnTC points out that the paper would have been stronger if it included theoretical exploration of the reasons behind the "importance of low SVs" phenomenon.

Reviewer Kuwu and jnTC consider the results marginally novel. Reviewer Kuwu considered the significance of the reported results to be limited, and put the work marginally above the acceptance threshold. Reviewer jnTC disagrees with this view, considers and appreciates the generality of the method and the fact that it can work well even for compressed models, while improving in accuracy by a few percent over competing approaches which result in similar parameter counts. The AC agrees with Reviewer jnTC.

Overall all reviewers consider the paper borderline but recommend accepting the paper. The AC overall the topic important (reducing the footprint of language models), the method simple and well motivated. The empirical evaluation is very thorough and shows clear gains across a large number of settings.